# The Roles of Post-Translational Modifications in STAT3 Biological Activities and Functions

**DOI:** 10.3390/biomedicines9080956

**Published:** 2021-08-04

**Authors:** Annachiara Tesoriere, Alberto Dinarello, Francesco Argenton

**Affiliations:** Dipartimento di Biologia, Università degli Studi di Padova, 35131 Padova, Italy; annachiara.tesoriere@phd.unipd.it (A.T.); alberto.dinarello@unipd.it (A.D.)

**Keywords:** STAT3, post-translational modifications, phosphorylation, acetylation, methylation

## Abstract

STAT3 is an important transcription factor that regulates cell growth and proliferation by regulating gene transcription of a plethora of genes. This protein also has many roles in cancer progression and several tumors such as prostate, lung, breast, and intestine cancers that are characterized by strong STAT3-dependent transcriptional activity. This protein is post-translationally modified in different ways according to cellular context and stimulus, and the same post-translational modification can have opposite effects in different cellular models. In this review, we describe the studies performed on the main modifications affecting the activity of STAT3: phosphorylation of tyrosine 705 and serine 727; acetylation of lysine 49, 87, 601, 615, 631, 685, 707, and 709; and methylation of lysine 49, 140, and 180. The extensive results obtained by different studies demonstrate that post-translational modifications drastically change STAT3 activities and that we need further analysis to properly elucidate all the functions of this multifaceted transcription factor.

## 1. Introduction

The signal transducer and activator of transcription (STAT) family is composed of seven members: STAT1, STAT2, STAT3, STAT4, STAT5a, STAT5b, and STAT6. These proteins work mainly as transcription factors that bind specific consensus palindromic sequences on DNA and share the same structure. STAT proteins are composed of the following: an N-terminal domain, which is responsible for dimerization, nuclear import, and tetramerization on target gene promoters with tandem binding sites; a coiled-coil domain containing the nuclear localization sequence (NLS) and consequently involved in nuclear import and export; a DNA binding domain (DBD) that binds promoters of target genes in consensus sequences; a linker region; a highly conserved SH2 domain involved in recognition of receptor subunits and in STATs dimerization by stabilization of STAT dimers; and a transactivation domain that mediates transcription of STAT target genes (Figure 1) [1,2]. The importance of these transcription factors in the control of cellular processes is known worldwide and has been confirmed by the strong conservation of *Stat* sequences in different eukaryotic organisms, from *Hydra magnipapillata* to *Caenorabditis elegans*, from *Dictyostelium discoideum* to *Drosophila melanogaster*, from *Danio rerio* to *Homo sapiens* [3]. Although all the STAT proteins have a pivotal role in cellular homeostasis, nowadays, the study of STAT3 has gained more importance because several functions of this protein have not yet been completely described and because it exerts many different functions in tumor growth and progression [4]. The canonical activity of STAT3 is triggered by the IL-6/JAK/STAT3 pathway which is induced after binding of interleukin (IL) 6 family member proteins (including IL-6, IL-11, ciliary neurotrophic factor (CNTF), leukemia inhibitory factor (LIF), oncostatin M (OSM), cardiotrophin 1 (CT1), cardiotrophin-like cytokine (CLC), and IL-27) with membrane receptors, which are tethered to GP130 transmembrane proteins. GP130s physically interact with Janus kinases (JAKs) in their cytosolic portion and the interaction between IL-6 family members with their cognate receptors determines conformational changes to the autophosphorylation of JAKs. Phosphorylated JAKs trigger the phosphorylation of tyrosine 705 (Y705) of STAT3 protein and pSTAT3 Y705 dimerizes and migrates to the nucleus, where it interacts with its responsive elements on DNA (STAT3 inducible elements, SIEs), inducing the transcription of target genes (Figure 2) [5]. Alternatively, STAT3 can also be phosphorylated in Y705 residue after binding of growth factors such as epithelial growth factor (EGF), transforming growth factor α (TGFα), platelet-derived growth factor (PDGF) and hepatocyte growth factor (HGF) with their transmembrane receptors [6,7]. STAT3 forms homodimers, however, under specific conditions it can also interact with STAT1, STAT5a, STAT5b, and STAT4, exerting both inflammatory and anti-inflammatory functions [8]. STAT3 is involved in many biological processes such as stem cell pluripotency maintenance by regulating DNA methylation [9]; wound healing and regeneration [10]; inflammation [11,12]; and metabolism [13]. Of note, the role of this protein in tumor progression has been extensively studied and several types of cancers are characterized by high levels of STAT3 protein and its persistent activation [4,14,15]. However, it is not yet clear whether these functions of STAT3 rely on its nuclear activities or are also controlled by its localization in other subcellular compartments. STAT3 has been detected in mitochondria and endoplasmic reticulum where it regulates mitochondrial DNA transcription, electron transport chain activity, and calcium homeostasis [16,17,18,19]; however, neither its mechanism(s) of translocation nor the mechanisms of action of the transcription factor in these organelles have, so far, been completely understood. Similar to other STATs, STAT3 undergoes several post-translational modifications (PTM) which regulate the different functions of this protein in the nucleus as well as in other subcellular compartments [20,21]. With this review, we sum up the discoveries about phosphorylation, acetylation, and methylation, highlighting the discrepancies among several studies and the importance of targeting these PTMs to therapeutically target STAT3 in cancer.

## 2. Phosphorylation

STAT3 undergoes several PTMs and the most studied is phosphorylation, consisting of the covalent addition of a phosphate group to specific amino acids. This PTM drastically modifies the properties and functions of protein. The canonical IL-6/JAK/STAT3 pathway relies on the phosphorylation of STAT3 in Y705 residue in the carboxy-terminus side of the chain, catalyzed by activated JAKs after cytokine stimulation. This modification leads to STAT3 dimerization and nuclear translocation, allowing STAT3 to act as a transcription factor. In the nucleus, STAT3 dimers interact with DNA on the SIEs and regulate the transcription of genes involved in stem cell pluripotency maintenance (*Klf4*), regeneration and wound healing (*Il10*), inflammation (*Il8rb* and *Cxcl2*), apoptosis (*caspase-3*), and many other important biological processes (Figure 2) [5,22,23,24,25]. 

Another phosphorylation site of STAT3 is the serine 727 (S727) residue. The mechanisms triggering this modification and the function of phosphorylated S727 are still debated. The kinases that are able to induce this modification are cell and stimuli specific [26] and they are mainly extracellular signal-regulated kinase (ERK) 1, ERK2, mitogen-activated protein kinase (MAPK) p38, c-Jun N-terminal kinase (JNK), and an H-7-sensitive kinase [27]. S727 phosphorylation has often been considered to be an enhancer of STAT3 nuclear transcriptional activities that probably acts by recruiting activating cofactors [27], such as NcoA [28] and CBP/p300 [29]. 

After the experiments of Boulton et al. [30] and of Zhang et al. [31], who discovered the STAT3 S727 phosphorylation on SDS-gel electrophoresis, Wen and collaborators [32] studied the functions of this PTM. They used COS monkey cells—characterized by low levels of *Stat3* transcript—and overexpressed a mutated form of Stat3 in which S727 had been replaced with an Alanine, hence, blocking the possibility of STAT3 being phosphorylated in the 727 position (Stat3 S727A) (Figure 3). Interestingly, an electrophoretic mobility shift assay (EMSA), used to determine the capability of protein to bind DNA, demonstrated that the S727A mutation did not affect the ability of STAT3 to bind SIE. Finally, the authors used U3A cells (that showed a lower response of endogenous STAT3 to IFN-α stimulation, but a higher response when new *Stat3* is introduced [33]) expressing *Stat3 S727A* to test its transcriptional activity with luciferase assay; the results confirmed that S727 phosphorylation was fundamental for maximal activation of STAT3 protein [32].

In 2000, Decker and Kovarik [34] noticed some contrasting results reviewing information available about the function of S727 phosphorylation, while Schuringa et al. [35] confirmed in hepatoma cells stimulated with IL-6s the results of Wen et al., and Kim and Baumann [36] reported that STAT3 wild type and STAT3 S727A activities on hepatoglobin acute phase promoter were similar. Decker and Kovarik concluded that the function of S727 phosphorylation depended on the specific target gene promoter and/or cellular context. In addition, Decker and Kovarik reviewed the studies about the interdependence of serine and tyrosine phosphorylations. Considering that, in cells transfected with *Stat3 S727A,* there is upregulation of Y705 phosphorylation [37], but also taking into consideration the possible role of S727 phosphorylation in enhancing STAT3-dependent transcription [32,35], they proposed that S727 phosphorylation negatively affected Y705 phosphorylation as a direct effect, but also activated mechanisms (probably cofactor recruitment) that could overcompensate the negative effect on Y705 phosphorylation. From the functional point of view, Decker and Kovarik [34] supposed that the phosphorylation of S727 was important for STAT3-dependent control of cellular growth and they reported that overexpressed STAT3 S727A had a dominant-negative effect on transformation mediated by v-Src [38], which was also reduced after inhibition of serine phosphorylation of STAT3 protein [39].

Further analyses on STAT3 S727 functions were carried out in vivo by Shen et al. [40], who generated mice with the S727A substitution in the *Stat3* gene (called SA allele). They used embryonic fibroblasts from homozygous *Stat3*^SA/SA^ mice, and they discovered a halved transcriptional response in mutants as compared with wild type fibroblasts, hence, confirming the results described in Wen et al., [32]. Since *Stat3*^SA/SA^ and *Stat3*^+/-^ mice did not show altered phenotypes, Shen and collaborators bred these two lines to generate *Stat3*^SA/-^ mice. This composite heterozygous line showed perinatal lethality (about 75% of the offspring died for no specified reasons) and lower embryonic and perinatal growth. The authors proposed a connection between this early phenotype and the altered levels of insulin-like growth factor (IGF-1) in the serum of newborn mice. According to the aforementioned data, fibroblasts from *Stat3*^SA/-^ foetuses showed no more than 25% of STAT3-dependent transcription after IL-6 stimulation. Finally, *Stat3*^SA/-^ mice showed a decreased level of thymocytes and an increased apoptosis, suggesting the presence of defects in the thymocyte survival mechanisms. Although this study did not dissect the functions of S727 phosphorylation in different tissues and the authors analyzed only fibroblasts from foetuses, the *Stat3*^SA/SA^ murine line represented a good tool to further investigate STAT3 S727 phosphorylation in an in vivo system.

More recently, Hazan-Halevy et al. [41] also discovered a cellular model to investigate the role of S727 phosphorylation. Although STAT3 Y705 is frequently constitutively phosphorylated in solid and hematologic tumors, they showed that in chronic lymphocytic leukemia (CLL), which is the most common leukemia in the Western hemisphere, STAT3 is constitutively phosphorylated in S727 residue rather than in Y705, which in turn can be transiently phosphorylated in this cellular model. In detail, Hazan-Halevy and collaborators demonstrated that S727 residue of STAT3 was constitutively phosphorylated in the peripheral leukemic blood cells (CD19+) from patients with CLL as compared with non-leukemic (CD19-) cells from the same patients and healthy donors. The authors also demonstrated that pSTAT3 S727 translocated into the nucleus and bound to DNA, regulating STAT3-target genes. pSTAT3 S727 was found in nuclear fractions of CLL cells and was also detected by confocal microscopy. In the nuclei of unstimulated CLL cells, pSTAT3 S727 interacted with SIEs, as revealed by EMSA and ChIP assays and activated the transcription of specific STAT3-dependent genes. According to these results, the binding of STAT3 to DNA was not affected by dephosphorylation in Y705 residue. Finally, they concluded that pSTAT3 S727 could be a specific marker and a therapeutic target for CLL [41].

As previously mentioned, phosphorylation of both Y705 and S727 has been widely correlated to cellular growth control and cancer onset. For example, in these years, it was demonstrated that progestins were able to activate STAT3 in breast cancers by inducing phosphorylation of Y705 or S727 [26,42]. Proietti and collaborators [42] showed that in murine (C4HD) and human (T-47D) breast cancer cells, treatment with the synthetic progestin medroxyprogesterone acetate (MPA) promoted phosphorylation of Y705 STAT3 and consequent nuclear translocation, SIE binding, and STAT3-dependent transcriptional activation mediated by JAK- and Src-dependent pathways. According to their results, the MPA treatment stimulated phosphorylation of STAT3 (Y705), JAK1, JAK2, and Src, and this effect was abolished by the progestin antagonist RU486, indicating the direct involvement of progestin receptor (PR) (demonstrated to be an interactor of STAT3 upon MPA stimulation). Notably, the phosphorylation of STAT3 on Y705 was completely abrogated after inhibition of JAK1, JAK2, and Src. Moreover, the authors demonstrated a correlation between progestin stimulation of breast cancer growth and progestin induction of pSTAT3 Y705 and its subsequent transcriptional activity. By transfecting C4HD cells with *Stat3C* and *Stat3 Y705F* (acting, respectively, as dominant active and dominant negative forms of *Stat3*) (Figure 3), Proietti and collaborators demonstrated that the MPA-stimulated growth depended on STAT3 phosphorylation and its consequent activity. The cells transfected with *Stat3 Y705F* showed lower growth rates and higher levels of apoptosis, suggesting that the dominant negative form of STAT3 determines growth inhibition by cell cycle arrest and apoptosis. In order to verify this effect in vivo, Proietti and collaborators inoculated the transfected cells in mice treated with MPA depot. Only a few mice injected with cells expressing *Stat3 Y705F* developed tumors and these tumors had a lower growth rate, confirming that targeting STAT3 (and phosphorylation of its residues) could be a very effective therapeutic strategy against breast cancer. 

According to Tkach et al. [26], treatment with the synthetic progestin MPA could also induce S727 STAT3 phosphorylation in murine (C4HD) and human (T-47D) breast cancer cells and this induction was mediated by the PR. The effect was abolished by progestin antagonist and *PR* gene knockdown (by siRNA) in C4HD cells or by stable *PR* knockout in T-47D cells (in which the inducibility was restored upon *PR* transfection). The authors hypothesized that S727 phosphorylation was induced by progestins through activation of the c-Src/p42/p44 MAPK pathway, inhibition of which (using either specific inhibitor or genetic mutations) blocks S727 phosphorylation after MPA stimulation. Tkach et al. also showed that phosphorylation of S727 increased STAT3 nuclear translocation and maximized STAT3 transcriptional activities. In detail, the S727 phosphorylation of STAT3 was fundamental for the full transcriptional activation of *cyclinD1* gene (a key cell cycle regulator in breast cancer) through pSTAT3 S727 recruitment on its promoter. Finally, they demonstrated that phosphorylation of S727 was needed for progestin-induced tumor growth, both in vitro and in vivo (through the injection of C4HD cells in BALB/c mice). The proliferation of the cells was reduced by transfection of *Stat3 S727A* vector in both murine (C4HD) and human (T-47D) breast cancer cell as compared with the control cells transfected with wild type *Stat3* or empty vector. Notably, mice also injected with C4HD cells transfected with *Stat3 S727A* vector showed a reduced growth of tumor, in which the levels of pSTAT3 S727 and of *cyclinD1* mRNA were lower as compared with the control [26].

In order to understand the role of STAT3 Y705 and S727 phosphorylation, Huang et al. [43] analyzed the different functions of these two PTMs in the fate choice of mouse embryonic stem cells (ESCs). These authors optimized an inducible system of STAT3 expression in *Stat3^-/-^* mESCs that could express either wild type *Stat3*, *Stat3 Y705F,* or *Stat3 S727A.* After inducing the different *Stat3* forms and LIF stimulation, they showed that *Stat3* expression was needed for cell survival and that Y705 phosphorylation was indispensable for the maintenance of pluripotency. In addition, their results showed that S727 phosphorylation was involved in cell survival and mitogenicity; cells expressing *Stat3 S727A* had reduced survival and proliferation rate as compared with cells expressing wild type *Stat3*. Moreover, also in mESCs, S727 was phosphorylated by ERK1/2 kinases. The role of S727 phosphorylation in the regulation of proliferation was probably connected to STAT3-dependent expression of *Myc* gene, which was abolished in mESCs expressing *STAT3 Y705F* and reduced in mESCs expressing *STAT3 S727A*. In addition, three other STAT3 target genes involved in the maintenance of pluripotency (*Socs2*, *Nanog*, and *Klf4*) had the same responsiveness of *Myc*, confirming the role of S727 phosphorylation as a transcription enhancer. The S727 phosphorylation also appeared to be involved in neuronal differentiation. Loss of pS727 resulted in a reduction in neuronal differentiation potential, recovered by the S727 phosphomimetic mutations, called *Stat3 S727D*, which substituted the serine with aspartic acid (Figure 3). Huang and coworkers also proposed an antagonistic role between S727 and Y705 phosphorylation in the reprogramming of epiblast-derived stem cells (EpiSC). The epiblast stem cell stage is an obligate transitional step for the mESC differentiation. During this stage, mESCs can be reprogrammed to naïve pluripotent stem cells by overexpression of STAT3. Huang and collaborators demonstrated that pSTAT3 S727 negatively affected this phenomenon. All in all, these authors suggested that a dynamic equilibrium between pS727 and pY705 determined fate decisions in mECSs through two distinct mechanisms, i.e., while Y705 phosphorylation is fundamental for self-renewal and pluripotency maintenance, S727 phosphorylation is involved in cell proliferation, survival, and pluripotency potential [43].

Specific interactors of STAT3 can affect its phosphorylation, altering the activities of the protein and leading to cell transformation. For example, in a study by Aziz et al. [44], they demonstrated that protein kinase Cε (PKCε) could interact with STAT3 in different types of human cancers. This protein has been previously defined as a transforming oncogene and a predictive biomarker for some types of human cancer. The STAT3-PKCε reciprocal immunoprecipitation has indicated this molecular interaction in skin melanomas, prostate, gliomas, bladder, colon, lung, pancreatic, and breast cancer cells [44,45,46,47]. Of note, the use of blocking peptide in the immunoprecipitation experiment inhibited this interaction, confirming the physical relation between PKCε and STAT3 suggested by Aziz and collaborators. According to their results, in some tumors (melanoma, glioma, pancreatic, and lung cancer cells) PKCε could induce phosphorylation of S727 residue leading to an increase in STAT3 transcriptional activity. The inhibition of PKCε by siRNA hampered, in turn, S727 phosphorylation (without affecting Y705 phosphorylation), STAT3 interaction with DNA, and STAT3-dependent gene expression. The effect of this inhibition resulted in a reduced invasiveness ability of cancer cells. PKCε silencing also reduced the activation of MAPK cascade, the inhibition of which further reduced PKCε-mediated S727 phosphorylation of STAT3. These data suggest that PKCε can mediate STAT3 S727 phosphorylation via MAPK cascade (RAF-1, MEK1/2, and ERK1/2) and that this mechanism is fundamental for the constitutive activation of STAT3 in human cancers mentioned above. Finally, the authors concluded that PKCε was an initial signal that induced STAT3 to sustain cancer invasiveness.

The state of STAT3 is also regulated by the protein tyrosine phosphatases (PTPs), which can dephosphorylate STAT3 residues. The mechanisms of PTP functions have only been partially clarified and are currently under investigation as possible targets for treatments against cancer [48]. 

The central role of STAT3 in the development of a large number of tumors makes this protein an attractive target for studies of cancer therapy. Starting from the observation that activated STAT3 is detected in a large fraction of lymphoid malignancies, Kuusanmaki et al. [49] looked for drugs that could inhibit WT STAT3, and also STAT3 isoforms with gain of function mutations (Y640F and D661V) leading to increased Y705 phosphorylation, as well as STAT3 dimerization and activation. They tested the drugs that could target STAT3 activity using different cell model systems (Ba/F3 cells, NK cell leukemia/lymphoma cells, and LGL leukemia patient samples) and identified four classes of drugs, among 306 approved compounds, that seemed to be effective against wild type STAT3 and its mutated isoforms, i.e., mTOR, JAK, Hsp90 and CDK inhibitors. After testing the different drugs in vivo, these authors concluded that JAK inhibitors could be an efficient in vivo therapeutic strategy because, even if they are less effective in cells expressing mutated forms of STAT3, they could inhibit microenvironmental cytokine stimulation and STAT3 hyperactivation even in STAT3-mutated malignancies. Nonetheless, the more promising treatments seemed to be Hsp90 inhibitors that exerted antitumoral functions both in cells expressing *Stat3 WT*, *Stat3 Y640F,* or *Stat3 D661V*. Finally, Kuusanmaki and collaborators concluded that Hsp90 inhibitors, also in combination with JAK or mTOR inhibitors, may be a very potent long-term therapeutic option for lymphoproliferative diseases characterized by STAT3 mutations.

After underlining the roles of both pY705 and pS727 of STAT3 in the regulation of nuclear activities, it is worthwhile mentioning that STAT3 can localize in different subcellular compartments, and exert functions that are independent from canonical STAT3 transcriptional activities [50,51,52]. Although the localization of STAT3 in mitochondria is still under investigation and its putative mechanism of mitochondrial translocation has not yet been described, Peron and collaborators [18] dissected the roles of Y705 and S727 phosphorylation in the mitochondrial activities of STAT3. In particular, using mESCs and zebrafish as an in vivo model, the authors demonstrated that STAT3 affected mitochondrial transcription and cell proliferation in mESCs and also in a specific pool of adult stem cells of zebrafish optic tectum. Using in situ hybridization, Peron and coworkers demonstrated that overexpression on murine *Stat3* mRNA in zebrafish determined an upregulation of both *mt_nd2* (used as a hallmark of mitochondrial transcription) and *pcna* (used as a marker of cell proliferation). Notably, the overexpression of *Stat3 Y705F* and *Stat3 S727A* did not stimulate expression of *mt_nd2* and *pcna*, suggesting that phosphorylation of both Y705 and S727 was required for increased transcription of these genes. Hence, the authors decided to analyze the effects of a mitochondrially targeted form of *Stat3* (named *MLS_Stat3_NES*) and demonstrated that phosphorylation of both Y705 and S727 was required for the correct activities of STAT3 in the mitochondrion. While Y705 appeared to be necessary for the proper localization of STAT3 in mitochondria, S727 was needed for the proper biological activities of this protein in the organelle. 

In summary, Y705 phosphorylation is involved in the STAT3 canonical activation mechanism leading to dimerization and nuclear translocation of protein [5] but this PTM is also necessary for STAT3 mitochondrial localization [18]. This modification is largely correlated to STAT3 function as a nuclear transcription factor regulating cell cycle, cell pluripotency, and cell proliferation [43]. As reviewed in Avalle et al. (2019), alterations in Y705 phosphorylation have been related to cancer onset and progression [42] and, consequently, pharmacological treatments that are able to target Y705 phosphorylation, for example, Hsp90 inhibitors [40], are considered to be promising for cancer treatment [49]. Regarding S727 phosphorylation, the literature does not allow one to completely clarify its biological role. According to the reported information, S727 phosphorylation is not involved in STAT3 DNA binding [32], but it is necessary for nuclear translocation [26] and maximal activation of transcription (evidence confirmed in vitro by Wen et al. [32] and in vivo by Shen et al. [40]) in specific cellular and promoter contexts [34] probably through recruitment of cofactors useful for transcriptional activation [28,29]. S727 phosphorylation is also fundamental for the biological function of STAT3 in mitochondria [18]. From a functional point of view, S727 phosphorylation is mainly important for STAT3-dependent cellular survival and proliferation [43] modulating key targets such as *Socs2*, *NanoG,* and *Klf4*. In correlation to these assigned functions, it is not surprising that S727 phosphorylation is involved in v-Src-mediated transformation [38,39] and in c-Src/p42/p44 MAPK pathway-dependent breast cancer onset [26]. In addition, STAT3-dependent PCKε oncogenic action detected in some solid tumors is also mediated by S727 phosphorylation [44]. In conclusion, this PTM also shows promising characteristics for cancer therapy.

The STAT3 protein can also be phosphorylated in other residues of transcriptional activation domain (TAD), in C-terminal of sequence: S691 [53], T714 [54,55], T717 [55], and S719 [56]. Although mass spectrometry analysis has revealed phosphate groups in the residues just mentioned, no clear results about their functions in the regulation of STAT3 activities have been provided yet. For this reason, in this review, we decided to focus our attention only on Y705 and S727.

## 3. Acetylation

Another fundamental post-translational modification that usually affects the activity of nuclear transcription factors is acetylation. Acetylation and deacetylation modulate the mechanisms of transcriptional activity by regulating the chromatin accessibility through histone modification, and the interaction of transcription factors with their responsive elements on DNA. Interestingly, acetylation of both histone and transcription factors are carried out by histone acetyltransferases (HATs), the enzyme that is able to transfer an acetyl group from acetyl-coenzyme A to the ε-amino group of lysine residues [57,58,59]. This modification is highly reversible and the balance between acetylation and deacetylation is regulated by histone deacetylases (HDACs) [60]. It has been demonstrated that STAT family members can be widely acetylated in different positions [61]. Several studies have demonstrated that STAT3 can be acetylated in both N-terminal (K49 and K87) and SH2 domain (K685 and K631) and the function of these modifications is currently debated. In 2005, Ray and collaborators (2005) identified K49 and K87 as direct targets of p300 (co-activator) acetyl transferase activity. According to their data, K49 and K87 acetylation stabilizes the STAT3–p300 interaction, which is required for transactivation of STAT3 target genes. In particular, Ray et al. investigated the importance of K49 and K87 acetylation for the transactivation of the human angiotensinogen (*hAGT*) gene, because STAT3 mediates the acute-phase response (APR) of the *hAGT* gene in hepatocytes. These authors demonstrated that in HepG2 cells, STAT3 was acetylated by the p300 co-activator upon IL-6 stimulation. They mapped K87 as the major and K49 as the minor Ac-acceptor site and they suggested that nuclear acetylation of these residues increased p300-STAT3 binding on the promoter regions of *hAGT* gene, hence, inducing its transcription. Moreover, Ray et al. [62] demonstrated the interaction of HDAC-1 with STAT3 and its effect on the trans activation of *hAGT*, proposing HDAC-1 as a negative modulator of this PTM. Further studies by the same research group [63] have better characterized the functions of K49 and K87 acetylation, i.e., both silencing and knock-out of *HDAC-1* determine accumulation of STAT3 in the nucleus and enhance STAT3-dependent transcriptional response after IL-6 stimulation. These results confirmed that acetylation of STAT3 was required for proper nuclear localization of the protein and to potentiate its transcriptional activities. 

Acetylation of K87 was also correlated to mitochondrial translocation of STAT3 by Xu et al. [13]. In particular, insulin-stimulated STAT3 was mostly acetylated and localized in organelle. Although this PTM enhances mitochondrial localization of STAT3, it is not indispensable for this process, indeed, the STAT3 N-terminal domain is responsible for a CBP-stimulated increase in mitochondrial translocation, and STAT3 N-terminal truncated form can also translocate in the organelle. Among all the mutations in the main STAT3 acetylation sites, only the K87R mutation (Figure 4) causes a reduction in mitochondrial translocation, even upon insulin stimulation. Possibly, the acetylation promotes mitochondrial translocation by neutralizing the lysine positive charge and, consequently, helps the protein to cross the negatively charged mitochondrial membrane. Finally, Xu et al. concluded that the balance between acetylated and deacetylated forms was maintained by HDAC6, which acted in the cytosol by removing the Ac group.

The SH2 domain of STAT3 can be acetylated. Ma and collaborators’ [64] spectrometric analysis showed that p300/CBP could acetylate multiple residues in the SH2 domain: K601, K615, K631, K685, and K707. Ma and co-authors identified these residues as targets of lysyl oxidase-like 3 (LOXL3) nuclear deacetylation and deacetylimination activities; therefore, LOXL3 can negatively modulate STAT3 dimerization and transcriptional activity. Considering that LOXL3 is downregulated in some types of cancer, they supposed that the acetylation of the SH2 domain affected the role of STAT3 in the regulation of the cell cycle. In addition, these authors also showed that LOXL3 expression was higher in the spleen and thymus suggesting a role of this protein in T cell development and differentiation [64]. 

The most studied acetylation of STAT3 C-terminal domain is on the K685 residue. In 2005, for the first time, Wang et al. [65] showed that, in mammalian cells, STAT3 could be acetylated in K685 by its coactivator CBP (p300/CREB-binding protein family). This modification was able to stimulate the sequence-specific binding of STAT3 to DNA and the transactivation of its target genes. They also demonstrated that the inhibition of histone deacetylase activity increased nuclear localization of STAT3. Wang and collaborators transiently overexpressed *Stat3* together with *CBP* or *p300* in HepG2 cells, showing that *STAT3 + p300* or *STAT3 + CBP* generated increased levels of acetylated STAT3. They mapped the STAT3 acetylation sites using in vitro acetylation assay and identified K685 as the target of p300. The interaction was confirmed in vivo using FLAG-tagged mutant proteins and performing a transient transfection acetylation assay. They also hypothesized that the acetylation of STAT3 in the nucleus prevents its nuclear export, increases its transcriptional activity, and recruits CBP/p300 on the promoter region of STAT3 target genes, activating their expression. They also supposed that the acetylation of K685 could influence STAT3 phosphorylation modulating the formation of STAT3 dimers. 

The hypothesis of K685 as a modulator of STAT3 dimer formation was also shared by Yuan et al. [66]. Their data suggested that K685 acetylation was critical for stable dimerization of STAT3, its cytokine-induced DNA binding, and transcriptional regulation. The point mutations of K685 residue (Figure 4) resulted in a lower transcriptional response (decreased expression of cyclin D1, Bcl-X_12_, and c-Myc proteins) to OSM stimulation, suggesting that acetylation of K685 changed the local charge of the protein and favored the formation of STAT3 homodimers. The same authors also showed that K685 acetylation was catalyzed by p300 and could be reversed by type I HDAC family members, even if the K685R mutation, which substituted Lysine with Arginine (Figure 4), did not determine detectable changes in the formation of complexes between STAT3 and either p300 or HDAC factors. Finally, Yuan et al. [66] demonstrated that STAT3 phosphorylation was not necessary for STAT3 acetylation, but concluded that Y705 phosphorylation and K685 acetylation alone seemed to be insufficient for STAT3 activation and that both of these PTMs were required to trigger STAT3-dependent transcriptional activities. 

The Yuan et al. study was further discussed in a study by O’Shea et al. [67]. The first point that O’Shea and collaborators contrasted was the role of K685 in STAT dimers formation, i.e., while K685 was close to the interface of interaction, structurally, it did not seem to be involved in the interaction. The second highlighted point was the effect of AcSTAT3 K685 on transcriptional activity. O’Shea and coworkers reported that the STAT response was blocked by inhibition or reduction of HDAC expression and suggested that the overexpression experiment in reporter lines only showed indirect effects. However, O’Shea and collaborators only cited experimental data collected on STAT1, STAT2, and STAT6 [68,69,70] and they extended their conclusions to STAT3, considering the strong conservation of the K685 residue in all these proteins. The next issue raised in the study by O’Shea et al. was about STAT3 interactions. Yuan et al. showed that after ligand stimulation, STAT3 was associated with p300 and dissociated from HDAC3. Because both proteins were in the nucleus as part of the multiprotein complex, O’Shea and collaborators claimed that there were not enough data to demonstrate a physiological relevance of the interactions proposed in Yuan et al. and the involvement of cytokines in this process needed to be further elucidated.

Despite all of the doubts expressed in the study by O’Shea et al. [67], the results of the study by Yuan et al. [66] have been taken into consideration by other research groups that wanted to study the function of K685 acetylation. Starting from the results by Yuan et al., Belo et al. [71] analyzed the effects of K685 acetylation on the STAT3 crystal structure and did not find direct effects of this modification on STAT3 DNA binding affinity or specificity, and they concluded that probably the STAT3 transcriptional activity identified as acetylation dependent in vivo was influenced by the specific cellular environment (other post-translational modifications, interaction with other proteins, and subcellular compartmentalization). Belo et al. analyzed the STAT3 core domain and hypothesized that N- or C-terminal regions of STAT3 had a role in the acetylation-dependent DNA interaction of STAT3. Using a deacetylation assay in bacteria, the authors identified sirtuin (SIRT) 1–3 and HDAC6 as enzymes that were able to hydrolyze AcK685. SIRT1 had previously been identified as a STAT3 modifier, while HDAC6 was a STAT3 core interactor [71]. Interestingly, SIRT3, which acts in mitochondria, can deacetylate STAT3 AcK707, AcK709 (targets previously identified by Xu et al. [13]), and AcK685 residues. SIRT1–3 and HDAC6 actions are dependent on Y705 phosphorylation state. The authors supposed that this dependence occurred because K685 was located in the dimer interface and the phosphorylation of Y705 promoted STAT3 dimerization.

A possible SIRT’s protein regulatory mechanism of STAT3 C-terminal domain acetylation was also suggested by Xu and collaborators [13] who supposed a connection between mitochondrial activity of STAT3 and the acetylated/deacetylated state of the protein. According to their results, inside mitochondria, STAT3 promoted the oxidation of pyruvate in acetyl-CoA by interacting with pyruvate dehydrogenase complex E1 (PDC E1, an upstream component of ETC), thus, elevating mitochondrial membrane potential and ATP production. The balance between acetylated and deacetylated forms of STAT3 seemed to be modulated by SIRT5 (a mitochondrial enzyme involved in different steps of TCA-ETC pathway), which could deacetylate STAT3, inhibiting its activity in mitochondria of HeLa cells. Xu et al. also identified SIRT3 as a modulator of STAT3 acetylation. SIRT3 and SIRT5 can both deacetylate K685, but only SIRT5 can deacetylate K707 and K709. Regarding the role of STAT3 in the regulation of mitochondrial metabolism, Xu and collaborators showed that CBP transfection elevated mitochondrial membrane potential and ATP production, both reduced by SIRT5 transfection. According to this evidence, the authors showed that PC3 cells (prostate cancer cell line bearing a STAT3 whole-gene-deletion mutation) had low mitochondrial membrane potential that increased significantly, together with ATP production, after *Stat3* transfection. Additionally, STAT3 was constitutively acetylated in mitochondria extracted from A549 lung adenocarcinoma cell line, where PDC E1 was overexpressed [13]. Xu et al. reported that, in cancer cells with a low Warburg effect (consisting in enhanced glucose conversion to lactate via pyruvate), for example, A549 lung cancer cell, STAT3 was constitutively acetylated in K685 and translocated into mitochondria (in steady state), where the SIRT5 level was lower than normal tissues, and the ATP production increased. The possible modulation of STAT3 mitochondrial activity by protein acetylation could be very interesting from a clinical point of view because the metabolic regulation of pyruvate conversion in mitochondria operated by STAT3 supports the Ras-dependent malignant transformation [72]. 

Interestingly, the acetylation of STAT3 protein could be correlated to the modulation of unphosphorylated STAT3 protein (concerning the Y705 residue). 

U-STAT3 has been supposed to regulate the transcription of a set of genes which are not targets of pSTAT3 homodimer. Yang et al. [73] previously identified a possible correlation between U-STAT3 action and oncogenesis. U-STAT3, whose expression represents a late response to IL-6 stimulation, mediated the transcription of some oncoproteins through a novel mechanism. This is important for understanding the response to IL-6 and also for clarifying the oncogenic mechanism in tumors characterized by a constitutively active form of STAT3 [73]. U-STAT3 seems to be also involved in cardiac hypertrophy. Yue et al. [74] identified that, in angiotensin II type 1 receptor (AT1R) transgenic mice, the expression of *Stat3* gene was induced promoting a persistent nuclear accumulation of U-STAT3. Interestingly, the authors correlated the accumulation of U-STAT3 in the nucleus to cardiac hypertrophy accompanied by pathogenic gene expression (upregulation of *Opn*, a marker for heart failure, and downregulation of *Rgs2*, a negative regulator of cardiac hypertrophy signaling).

Interestingly, Dasgupta et al. [75] reported that U-STAT3 transcriptional activity stimulated by angiotensin II relied on K685 integrity. Specifically, U-STAT3 acetylation on K685 was important for the expression of a large set of genes (more than 70%) regulated by U-STAT3 in hTERT-HME1 cells. These authors also confirmed, in accordance with Yuan et al. [66], that mutation in K685 did not influence Y705 phosphorylation or IL-6-stimulated gene expression in human prostate cancer cells PC3. According to Yue et al. [74], Dasgupta and collaborators identified that K685 was fundamental for angiotensin II-dependent gene expression of some target genes. Angiotensin II type 1 receptor activation was correlated to U-STAT3 nuclear accumulation and promoted U-STAT3 interaction with p300 which led to the expression of genes involved in cardiac hypertrophy and dysfunction [75].

As previously mentioned, STAT3 acetylation can have a pivotal role in the regulation of tumorigenesis. Lee et al. [76] reported that K685 was highly acetylated in melanoma tissue, colon cancer, and triple-negative breast cancer. The authors proposed that, in the mentioned tumors, STAT3 interacted with DNA methyltransferase 1 (DNMT1) and the STAT3-DNMT1 complex catalyzed the methylation and consequent inactivation of some tumor-suppressor genes such as *p53* and *PTPN6.* Interestingly, the STAT3 K685R mutation reduced tumor growth through interruption of the STAT3–DNMT1 interaction and consequent reactivation of a set of tumor-suppressor genes. Hence, Lee et al. suggested that K685 acetylation helped STAT3 and DNMT1 to colocalize properly in tumor cells, methylate promoter regions of tumor-suppressor genes, and inhibited their expression. Among these genes, the estrogen receptor alpha (*ERα*) could be reactivated in triple breast cancer cells by using resveratrol, a non-flavonoid phenol that is able to dampen K685 acetylation. Moreover, the chromatin modification in the *ERα* promoter region predicted melanoma progression and the reactivation of this gene made tumor cells sensitive to antiestrogens (that normally cannot be used to treat patients affected also by colorectal cancer, lung cancer, and melanoma in which the *ERα* promoter region is methylated). Additionally, Lee et al. confirmed that the K685 mutation had a small effect on STAT3 phosphorylation [76]. 

In addition, Gupta et al. [77] studied the role of STAT3 acetylation in cancer and in large B-cell lymphoma (DLBCL). They took into consideration two specific DLBCL forms, i.e., the still poorly understood activated B-cell-like (ABC) type and the germinal center B-cell-like (GBC) type. They used human samples and DLBCL cell lines and identified that both HDAC3 and pSTAT3 Y705 were aberrantly co-expressed and interacted with each other in ABC DLBCL cells. By analyzing the HDAC3 role on the acetylation of STAT3 K685 in malignant lymphoma, the authors showed that HDAC negatively regulated this PTM, i.e., *HDAC* overexpression decreased AcSTAT3 K685 levels, while *HDAC* knockdown upregulated acetylation of STAT3. Finally, they demonstrated that the HDAC inhibitor panobinostat (LBH589, used in clinical trials for lymphoma and myeloma) was able to increase the acetylation level of endogenous STAT3 in a dose-dependent manner. In agreement with Wang et al. [65], Gupta and collaborators demonstrated that p300 and STAT3 physically interacted with each other and panobinostat treatment increased the level of acetylated STAT3 associated with p300, inducing cell death through downregulation of *Mcl-1* and *c-Myc* expression and an increase in PARP cleavage. Additionally, their data suggested that the IL-10 stimulation of STAT3 nuclear translocation was reduced by panobinostat treatment, pointing to a role of HDAC in mediating nuclear translocation. This compound dephosphorylates STAT3 Y705 in a dose dependent manner by the JAK2 independent mechanism. All in all, panobinostat increases STAT3 acetylation and results in decreased STAT3 accumulation in the nucleus along with a significant decrease in nuclear pSTAT3 Y705. Since K685 and Y705 residue are close to each other, Gupta et al. supposed that panobinostat-induced K685 acetylation prevented tyrosine phosphorylation. However, it is worthwhile noting that the results shown by Gupta and collaborators are in contrast to the observations by Wang et al. [65] and Yuan et al. [66], suggesting that the relationships between STAT3 post-translational modifications need to be further elucidated and that they have different effects in different models and tissues. 

HDAC inhibitors have also been identified as possible pharmacological treatments by Sun et al. [78], who hypothesized that STAT3 acetylation was critical for induction of indoleamine 2,3-dioxygenase (IDO) in dendritic cells (DC) and suggested HDAC inhibitors as modulators of this process. These authors also confirmed the presence of the STAT3/p300 complex in DC through immunoprecipitation and determined that HDAC inhibitors promoted STAT3 dimers formation, in contrast to the observations by Gupta and collaborators [77]. Hence, Sun et al. hypothesized that HDAC was able to block p300 activity in the STAT3/p300 complex and that HDAC inhibitors could reverse this process. They also demonstrated that the inhibition of STAT3 binding to DNA through JSI-124, a JAK inhibitor [79] treatment, abrogated the *IDO* mRNA elicited by HDAC inhibition; the direct link between *IDO* transcription and STAT3 action was confirmed by the discovery of STAT3 consensus elements in *IDO* gene promoter sequence which contained two gamma interferon activation (GAS) sites corresponding to STAT3 binding consensus elements on DNA. To demonstrate the direct link between STAT3 binding to the *IDO* promoter region and HDAC modulation activity on *IDO*, Sun and collaborators performed luciferase reporter assays using a mutated *IDO* promoter region. The results showed that *IDO* transcription was stimulated by treatment with SAHA (a HDAC inhibitor), but that this stimulation was blocked when the promoter region was mutated in the GAS. This induction was also blocked by JSI-124 treatment, demonstrating that both STAT3 phosphorylation and acetylation could affect the STAT3-dependent transcription of *IDO*. Furthermore, since K685R mutation blocked the SAHA-dependent promotion of *IDO* transcription in transfected cells, Sun and collaborators demonstrated a direct correlation between *IDO* transcriptional stimulation and the K685 acetylation induced by HDAC inhibition [78].

Another molecule of pharmaceutical interest targeting STAT3 post-transcriptional modification is garcinol. This polyisoprenylated benzophenone extracted from *Garcinia indica* was used by Sethi and collaborators [80] as an inhibitor of STAT3 acetylation and as a possible treatment for human hepatocellular carcinoma (HCC). STAT3 is constitutively activated and highly acetylated in the HCC and contributes to tumor progression by inducing the methylation of tumor suppressor genes. Of note, garcinol treatment results in a reduction of AcSTAT3 levels through inhibition of p300 acetyltransferase activity. Sethi and collaborators showed that garcinol suppressed both constitutive and inducible STAT3 activation by inhibiting acetylation and JAK2-dependent tyrosine phosphorylation. Consequently, garcinol blocks STAT3 dimerization, probably by interacting directly with the SH2 domain. Docking prediction and subsequent in vitro and in vivo tests have confirmed that a reduction in STAT3 dimerization depends on garcinol physical interaction with STAT3. A reduction in STAT3 dimer formation also impairs the nuclear localization of this transcription factor and its interaction with DNA. On the one hand, after garcinol treatment, in vitro assays show a significantly reduced ability of STAT3 to bind DNA. On the other hand, garcinol strengthens STAT3 monomer interaction with DNA on *c-fos* promoter. Additionally, garcinol reduces HCC cells survival, suppresses proliferation, and increases apoptosis by downregulating the expression of STAT3-target genes involved in proliferation (*cyclin D1*), apoptosis (*Bcl-2*, *Bcl-xL*, *survivine*, and *Mcl-1*), and angiogenesis (*VEGF*). Garcinol can also act in combination with other anticancer agents to enhance apoptosis and retard cellular proliferation in HCC. Finally, garcinol effects were also recapitulated in athymic mouse models bearing hepatic cancer xenografts. In particular, intraperitoneal injection of garcinol in these mice resulted in inhibition of STAT3 and a reduction in tumor growth [80].

On the basis of the information reported in the literature, we can conclude that the main STAT3 acetylation sites are K87 and K685. K87 acetylation is mediated by p300 and promotes STAT3–p300 interaction leading to transactivation of STAT3 target genes [62]. This modification is involved in STAT3 nuclear localization [63] and has an accessory role in STAT3 mitochondrial translocation [13]. In addition, K685 acetylation is catalyzed by CBP/p300 and is related to STAT3 nuclear localization (probably by preventing STAT3 nuclear export) and transactivation of STAT3 targets [65]. Although the role of acetylation in the induction of STAT3 nuclear transcriptional activity has been demonstrated [73,74,75,78], the relationship between K685 acetylation and the Y705 phosphorylation is not completely understood. Y705 phosphorylation is not needed for K685 acetylation [66] since AcSTAT3 K685 characterizes also U-STAT3 in hTERT-HME1 cells [75], but both modifications seem to be necessary for STAT3-dependent transcriptional activity [66]. Moreover, the K685 position is close to Y705 and can affect its phosphorylation. Although O’Shea et al. [67] suggested that K685 should not be structurally involved in the STAT3 dimerization interface, many studies have reported that the local charge change induced by K685 acetylation could promote STAT3 dimerization [65,66], independent of the specific cellular environment [71]. At the same time, Gupta et al. [77] proposed that K685 acetylation could prevent Y705 phosphorylation, while Belo et al. [71] showed that Y705 phosphorylation protected K685 against deacetylation by HDAC6, SIRT1, and SIRT3 (the latter enzyme acts in mitochondria, where K685 acetylation seems to regulate STAT3 metabolic functions [63]). Experimental evidence has demonstrated that AcSTAT3 K685 relies on IL6 stimulation [81], hence, STAT3 acetylation, which enhances STAT3 activities, can be involved in a signaling loop that fuels the IL-6/JAK/STAT3 pathway. In addition, the aforementioned results demonstrate that acetylation of STAT3 has essential roles in tumor growth and immune response regulation. Since the effects of STAT3 acetylation are different in different pathologies [76,77,78,80], the scientific community is investigating possible chemical inhibitors of histone deacetylases and STAT3 acetylation as promising pharmacological treatments. On the one hand, studies have shown chemical compounds that are able to inhibit K685 acetylation (e.g., resveratrol [76] and garcinol [80]), which is correlated with STAT3-dependent methylation of tumor suppressor genes; on the other hand, studies have tried to find HDAC inhibitors (e.g., panobinostat [77] and SAHA [78]) for the treatment of malignant lymphoma (in which HDAC inhibition promotes cell death) [77] or to regulate the immune response (by modulating *IDO* induction in dendritic cells) [78].

## 4. Methylation

STAT3 can be methylated in both its N-terminal domain (K49) and in adjacent coiled-coil domain (K140 and K180). Enhancer of zeste 2 polycomb repressive complex 2 (EZH2) is the histone-modifying enzyme identified as being responsible for K49 di-methylation and K180 methylation. Dasgupta et al. [82] suggested that EZH2 acts on K49 residue after Y705 phosphorylation. In particular, these authors demonstrated that K49 di-methylation is essential for the expression of STAT3-dependent genes upon IL-6 stimulation: indeed, the K49R (Figure 5) mutation of STAT3 significantly downregulates the expression of a large subset of genes (33 out of 59) that are normally induced by IL-6 stimulation. Additionally, Dasgupta and collaborators suggested that STAT3 methylation relied on pSTAT3 Y705, demonstrating that canonical activation of STAT3 potentiates its effects by inducing methylation. In addition, Kim et al. [83] reported that EZH2 K180 methylation promotes STAT3 Y705 phosphorylation. These authors showed that EZH2, after the specific S21 phosphorylation by Akt, binds STAT3 and methylates it, hence promoting Y705 phosphorylation through a not well specified mechanism. Kim et al. supposed that MeSTAT3 K180 enhanced Y705 phosphorylation by protecting this residue from dephosphorylation. They reported that both *EZH2* knockdown and EZH2 inhibition (with DZNep) caused a decrease of pSTAT3 Y705 levels. These data were confirmed also by a reduction in the expression levels of validated STAT3 target genes (like *SOCS3* and *c-MYC*), as well as a reduced STAT3 reporter activity, upon DZNep treatment. The EZH2 role in the modulation of STAT3 activities has a significant clinical relevance mostly because EZH2 overexpression determines pro-tumorigenic effects in solid cancers, in which STAT3 is often constitutively activated [82]. The STAT3-EZH2 interaction seems to be fundamental in glioblastoma multiforme stem-like cells (GSCs), being responsible for glioblastoma multiforme (GBM) spread, therapy resistance, and relapse [83]; Kim and collaborators, using GSCs, demonstrated that EZH2 induced transcriptional silencing, and also activated STAT3. In GBM cells, the overexpression of EZH2 correlated with the worst prognosis, as well as STAT3 hyperactivation (which is a common feature of many tumors). These results drove other research groups to test some highly specific EZH2 inhibitors as potential therapeutic agents [84,85,86]. Notably, the physical interaction between STAT3 and EZH2 has recently been studied in *Xenopus laevis*. Loreti and collaborators demonstrated that STAT3-EZH2 interaction resulted in methylation of STAT3 and in the consequent MeSTAT3-dependent regulation of dorso-ventral patterning during *X. laevis* development [87]. 

Interestingly, methylation of STAT3 on K180 and K49 residues have an opposite effect on the regulation of STAT3-dependent transcriptional activity as compared with K140 STAT3 di-methylation, which negatively regulates STAT3 dependent transcription. Yang et al. [88] identified that histone-lysine N-methyltransferase SET9 was responsible for di-methylation of K140, while lysine-specific demethylase 1 (LSD1) was able to demethylate MeSTAT3 K140. After IL-6 stimulation and consequent phosphorylation of STAT3 Y705 and S727, SET9 di-methylated K140. Yang and collaborators demonstrated that pSTAT3 S727 was needed for K140 di-methylation, hence, they supposed its involvement in a possible SET9–STAT3 interaction. As previously reported, this methylation is a negative regulator for the expression of a specific subset of STAT3 target genes. Indeed, Yang et al. identified three classes of genes that differentially responded to the absence of K140 methylation. Some STAT3 target genes (such as *SOCS3*, *FGF21*, *IRF8*, and *IRF9*) were upregulated in STAT3 K140R cells as compared with wild type cells, while other genes (such as *HSF1* and *CDCA1*) were downregulated; conversely, another group of genes (such as *CD14)* did not show expression alterations. 

All in all, the evidence mentioned thus far has demonstrated that methylation is fundamental for the regulation of STAT3 activity. MeSTAT3 K49, on the one hand, is induced by pSTAT3 Y705 and potentiates STAT3 transcriptional activity [82]; MeSTAT3 K180, on the other hand, induces pSTAT3 Y705 activating the pathway [83]; and MeSTAT3 K140 negatively regulates a subset of STAT3 target genes [79]. Since STAT3 has a prominent role in tumor growth and progression, the results described above about STAT3 methylation must be considered in studies on STAT3 inhibitors as therapeutic treatments.

## 5. Discussion and Conclusions

In this review, we have described the most relevant PTMs of STAT3 which can significantly affect both the canonical and the non-canonical activities of this protein. First, we focused on the activities and functions of phosphorylation that were better described in the literature. The Y705 phosphorylation is required for canonical nuclear activity of STAT3, i.e., it appears to be indispensable for the correct dimerization of the protein and the interaction with STAT3 binding elements. The induction of most target genes cannot be triggered if Y705 is not phosphorylated. As a consequence, the accumulation of U-STAT3 in the nucleus seems to be a late response to IL-6/JAK/STAT3 axis stimulation [73,89]. Tyrosine phosphorylation also regulates other PTMs of STAT3, such as acetylation of K685 and methylation of K180 which, in turn, enhance specific outcomes of pathway activity. Other PTMs strengthen the STAT3-dependent transcriptional activity both inducing pSTAT3 Y705 or Y705-independent mechanisms, affecting the expression of specific subsets of genes. Additionally, some PTMs affect the subcellular localization of STAT3, leading to the induction of non-canonical STAT3 functions in the endoplasmic reticulum and in mitochondria. Different STAT3 isoforms that have previously been tested for blocking or mimicking protein PTMs are currently available (Figure 3, Figure 4 and Figure 5). The constructs coding for these mutated forms of STAT3 are also mainly used in most cited studies, because they represent simple and useful tools to investigate the different functions of STAT3 PTMs. However, it is very important to note that precise characterization of PTM function is not trivial. The data shown in this review describe phosphorylation, acetylation, and methylation in STAT3 residues as regulators of the protein functions, but a tissue-specific analysis of each PTM effect on STAT3 activity is necessary to understand the different regulation mechanisms. It is tempting to speculate that most of the mentioned PTMs are indeed working as cytokine, cross-dependent synergistic mechanisms that irreversibly lead to progressive activation of STAT3 target genes. 

It is worthwhile mentioning that STAT3 can interact with several different proteins such as glucocorticoid receptor (GR) [90], androgen receptor (AR) [91], PR [26,42], hypoxia inducible factor (HIF) 1α [92], NF-κB [93], and many other molecules that can both inhibit and activate STAT3 functions [94,95]. Moreover, STAT3 can form heterodimers with other members of the STAT family [8]. All these STAT3 interactors are not ubiquitously expressed and can affect STAT3 activities and PTMs in a tissue- and stimulus-specific way. The biochemical and consequent cell-specific function of STAT3 PTMs will be ultimately dissected or not depending on the abilities of studies to separate STAT3 specific activities from those of other STATs. In different cellular backgrounds in which wild type STAT5 and STAT1 can be phosphorylated and heterodimerize with mutant STAT3, it may be impossible to distinguish the different contributions conveyed to homo- and heterodimers by the different members. For this reason, the study of STAT3 functions with genetic models such as zebrafish [95], easily targeted using CRISPR/Cas9, can significantly increase our knowledge about STAT3 PTMs functions both in the regulation of STAT3 activities and in the interaction with other proteins. Additionally, as recently observed by Grillo et al. [96] and Meier et al. [97], the cellular status and health can severely affect STAT3 functions. An increase in reactive oxygen species (ROS), which are generally produced by mitochondrial-related processes, reduces the levels of STAT3 in mitochondria [96], while having relevant roles in the nuclear activities of this protein. In high ROS conditions, cysteine residues of STAT3 DBD can be oxidized affecting the ability of the protein to interact with SIEs. In particular, Grillo and collaborators demonstrated that, one the one hand, upon ROS stimulation, genes involved in cell-cell adhesion, immune response, and transport were upregulated by STAT3; on the other hand, the STAT3-dependent expression of genes regulating tissue development and morphogenesis were downregulated by oxidized STAT3 [96]. Since elevated levels of ROS represent a hallmark of tumor growth and progression [98], oxidized STAT3-dependent transcriptional regulation can have an impact in these processes and should be further analyzed in the future. Hence, an attempt to generalize, and consequently to simplify, the information about the mechanisms used by cells to regulate STAT3 actions can lead to paradoxes and contradictions. For these reasons, our approach, in this review, focused on single residue modifications, evaluating the expression readouts and reporting the only apparently contradictory results. However, we highlighted the fact that the effects of PTM in STAT3 could lead to different results, according to the cellular context in which STAT3 was working. Hence, the same PTM might have different effects in different cells and only the tissue-specific analysis of STAT3 PTM can help to characterize STAT3 regulation and function in many pathologies and, in particular, in tumors, and thus allow one to identify more promising and specific treatments.

## Figures and Tables

**Figure 1 biomedicines-09-00956-f001:**
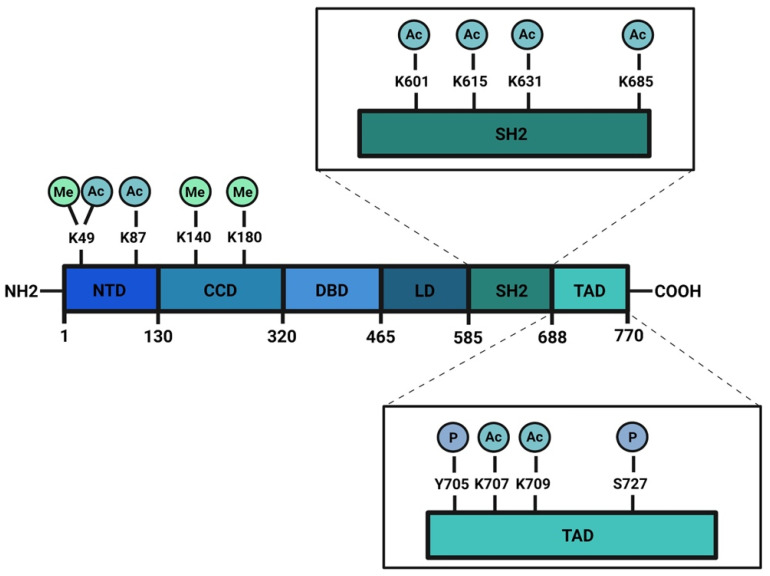
Schematic structure of STAT3 protein: STAT3 has a N-terminus domain (NTD), a coil-coiled domain (CDD), a DNA binding domain (DBD), a linker region (LD), a SH2 domain, and a transactivation domain (TAD) in the C-terminal portion of the protein. Phosphorylation, acetylation, and methylation sites are highlighted. Created with BioRender.com.

**Figure 2 biomedicines-09-00956-f002:**
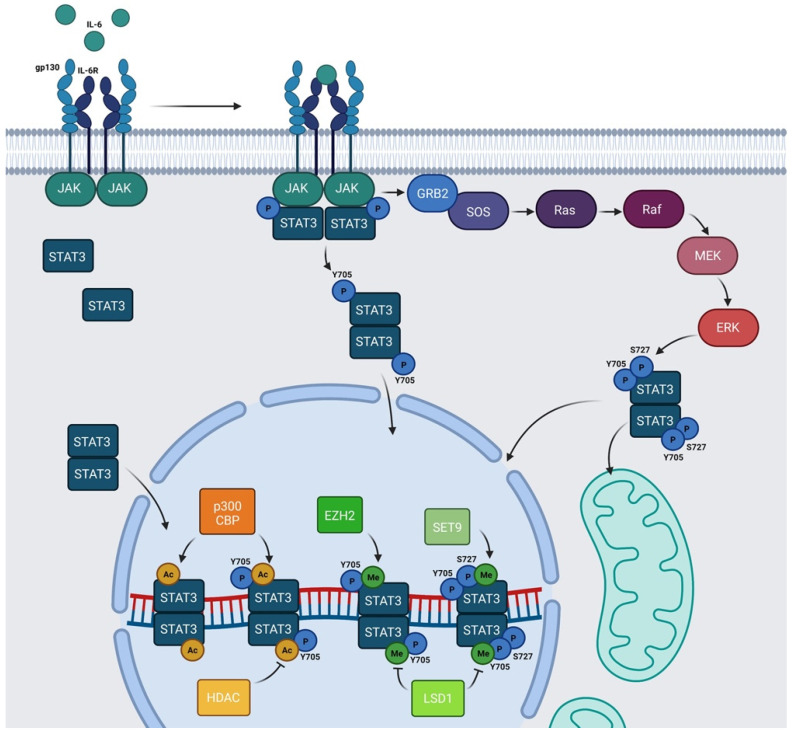
Schematic representation of IL-6/JAK/STAT3 pathway: IL-6 family members recognize their cognate transmembrane receptors and transmembrane proteins associated with the receptors trigger the activation of JAKs. Activated JAKs phosphorylate Y705 of STAT3 and pSTAT3 Y705 migrates to the nucleus. In the nucleus, STAT3 can be acetylated by p300/CBP and methylated by SET9 or by EZH2. HDAC and LSD1 inhibit AcSTAT3 and MeSTAT3, respectively. The Ras/Raf/MEK/ERK pathway determines S727 phosphorylation. Y705 and S727 are both required for mitochondrial functions of STAT3. Alternatively, unphosphorylated STAT3 (U-STAT3) can also regulate specific nuclear targets. Created with BioRender.com.

**Figure 3 biomedicines-09-00956-f003:**
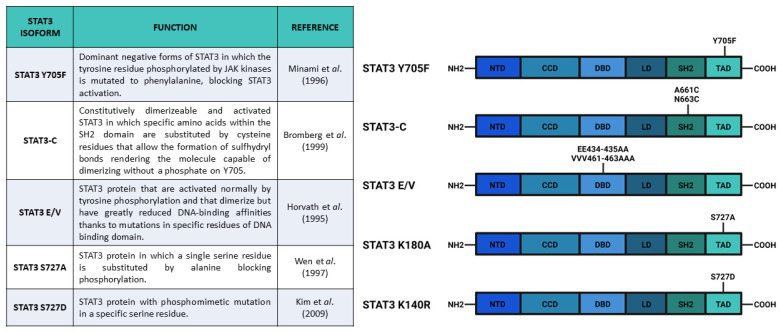
Main constructs used to investigate STAT3 phosphorylation. Created with BioRender.com.

**Figure 4 biomedicines-09-00956-f004:**
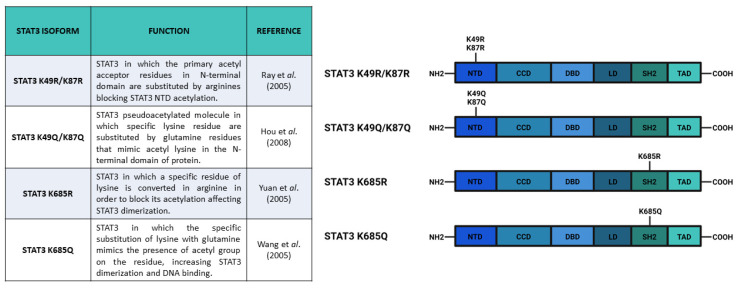
Main constructs used to investigate STAT3 acetylation. Created with BioRender.com.

**Figure 5 biomedicines-09-00956-f005:**
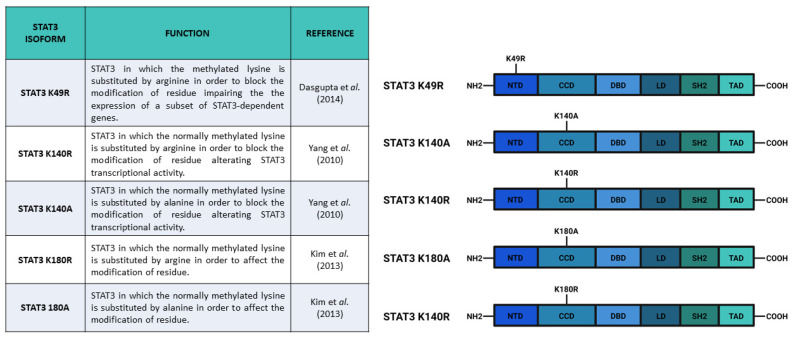
Main constructs used to investigate STAT3 methylations. Created with BioRender.com.

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
