# Peer review of "The Roles of Post-Translational Modifications in STAT3 Biological Activities and Functions"

_biomedicines, 2021, doi:10.3390/biomedicines9080956_

Round 1

Reviewer 1 Report

Thank you for the opportunity to review this paper. It has long been my opinion that a thorough review of STAT3 PTMs would be valuable for the field and I appreciate the effort hat went into this manuscript.  Unfortunately, I do not think that the format of this review ads to this field of work.  It feels throughout that the authors have chosen to largely summarize papers on different PTMs in separate paragraphs and have avoided discussing the conflicting and ambiguous data around pSTAT3 727, STAT3 acetylation and methylation. This would be of great value to the field and would allow readers to gain insight on how these papers relate to each other, otherwise, they should just refer to those papers and read them if no other commentary is given in the context of this review article. There has been good summarization of the work on STAT3 PTMs here, but the real value of this (in my opinion) would be comparing, contrasting and commenting on how these studies fit together. I would suggest major revisions to the format in order to bring this up to publication standards. 

Author Response

We have taken seriously the criticism raised by the referee and discussed in many more details the connections between different PTM and contrasting literature.  The literature is expanded (we can't say complete...) and we went through the text to eliminate all typos and grammatical flaws. In particular, discussion has been quite expanded.

Reviewer 2 Report

The review on Role of post translational modifications in STAT3 biological activities and functions is very interesting and is documented by a large number of references. However, I think that to make it easier to understand, in the sections in which the work is divided, in vitro studies with cells should be differentiated from in vivo studies. Likewise, a scheme could be made showing the changes in STAT3 activities in relation to different diseases. 

Author Response

We appreciate the referee suggestions but we still think that , whether studied in cell or in vivo, is better characterised if all the experimental data of each specific residue PTM are considered together to give an integrated view of its potential function. 

Round 2

Reviewer 1 Report

Thank you for addressing the previous concerns, this update version is much improved and the summaries are a very valuable aspect of review papers.